# Conversion of Enantiomers during the Separation of Acetoin from Fermentation Broth

**Jiaxiang Zhang [1,2,3,\*], Zhihao Fu [1], Xiangying Zhao [1,2,3,\*], Mingjing Yao [2,3], Yuchen Li [1], Liping Liu [2,3], Jianjun Liu [1,2,3] and Yanjun Tian [1,2,3]**

[1] School of Bioengineering, Qilu University of Technology (Shandong Academy of Sciences), Jinan 250306, China; 18505476551@163.com (Z.F.); lychen1735@163.com (Y.L.); liujj-2000@163.com (J.L.); tianyanjun16@163.com (Y.T.)

[2] Shandong Food Ferment Industry Research & Design Institute, Qilu University of Technology (Shandong Academy of Sciences), Jinan 250013, China; mingjing1004@163.com (M.Y.); spyllp@126.com (L.L.)

[3] School of Food Science and Engineering, Qilu University of Technology (Shandong Academy of Sciences), Jinan 250013, China

\* Correspondence: jxzh23@163.com (J.Z.); xyzhao68@126.com (X.Z.); Tel.: +86-150-6339-6995 (X.Z.)

**Abstract:** Acetoin (AC) is an important platform compound with two enantiomers (*R*)-AC and (*S*)-AC. Due to its unique spatial structure, optically pure AC has particularly high application in asymmetric synthesis. Highly optically pure AC could be produced from glucose using biofermentation technology. In this paper, we have observed that the recovered AC product from the fermentation broth containing (*R*)-AC was a racemic mixture. The changes of the enantiomeric excess (e.e.) of (*R*)-AC enantiomers in the feed solution during the recovery process were then investigated, confirming that the racemization occurs during solvent distillation. Further studies showed that high temperature is the main factor affecting the conversion of the two enantiomers, while low temperature significantly prevents this conversion reaction. Therefore, we optimized the solvent recovery process and used vacuum distillation to reduce the distillation process temperature, which effectively prevented the racemization: obtains AC products with more than 98% purity and successfully maintained the proportion of (*R*)-AC above 96%. To our knowledge, this is the first report on the factors affecting the enantiomeric purity in the downstream extraction process of AC production by fermentation.

**Keywords:** acetoin; fermentation broth; downstream; temperature; optical purity

## 1. Introduction

Acetoin (3-hydroxy-2-butanone, AC) is an important chiral substance [1] with a stereogenic center within its four-carbon molecule (Figure 1); it has widespread applications in food [1], chemical industry [2], medicine [3], and other fields. In particular, optically pure AC not only possesses the basic functional properties of AC but also offers a prominent advantage in asymmetric synthesis owing to its unique spatial structure. This feature confers special application value in the synthesis of high-value-added chiral drug intermediates, chemical intermediates, and liquid crystal materials [4,5].

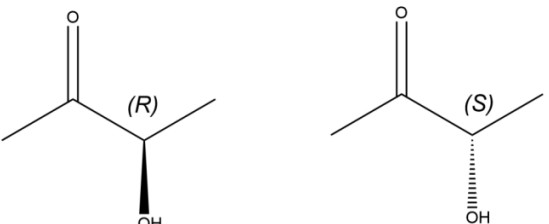

**Figure 1.** Configurations of the two enantiomers of AC.

Currently, AC is mainly manufactured through chemical methods, yielding racemic mixtures of (R)-AC and (S)-AC. This racemic synthesis limits the application of chemically generated AC [6–8]. Given that microorganisms contain a specific 2,3-butanediol dehydrogenase (BDH), high optical purity AC can be produced by biotechnology [9,10]. However, most AC products obtained by microbial fermentation are also a mixture of (R)-AC and (S)-AC [11], which is due to the presence of various types of BDH in bacteria. The AC synthesis pathway in bacteria comprises the following steps (Figure 2): conversion of α-acetolactate to (R)-AC under the catalytic action of α-acetolactate decarboxylase (ALDC) and conversion of (R)-AC to (R,R)-2,3-butanediol (R,R)-2,3-BD) and *meso*-2,3-BD under the catalysis of different types of BDH [12,13]. The aforementioned conversion reactions of AC to BD catalyzed by BDH are reversible; therefore, the conversion of *meso*-2,3-BD to (S)-AC may occur. In general, bacteria contain more than one type of BDH [14,15], thereby producing both AC enantiomers. To obtain optically pure AC (i.e., only one AC enantiomer), researchers have used various approaches, including the screening of natural bacterial strains for the synthesis of AC with high optical purity [16], engineering strains that possess BDH with strict enantioselectivity [17,18], and deletion of *bdhA* gene in AC-producing bacteria, to block the conversion between AC and BD [19]. Several studies have demonstrated the effectiveness of these strategies and the optical purity of AC in the product can be increased to more than 98% [16,19]. In addition, chiral pure AC has been manufactured by enzymatic [20,21] and whole-cell methods [22].

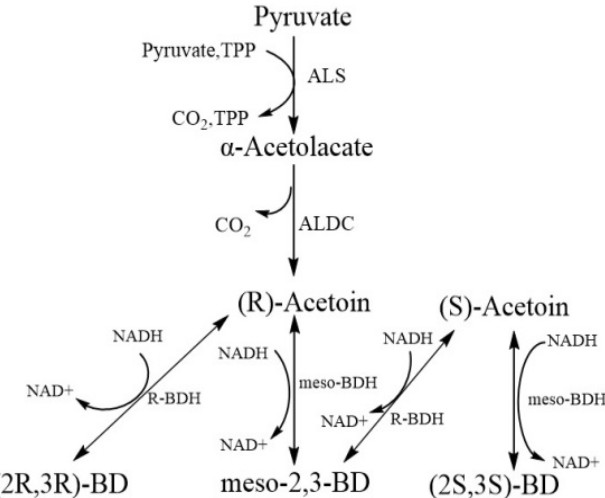

**Figure 2.** Metabolic pathways of AC in organisms, adapted with permission from Ref. [23]. ALS: Acetolactate synthase; ALDC: Acetolactate decarboxylase; R-BDH: R-2.3-Butanediol dehydrogenase; meso-BDH: meso-2.3-Butanediol dehydrogenase.

The downstream isolation of AC is important after optimizing the metabolic mechanism of microorganisms and fermentation conditions. Salting-out [24] and aqueous two-phase extraction [25] are currently the main methods used to extract AC from the fermentation broth; the authors' research group added 25% $Na_2SO_4$ to the filtrated fermentation broth and extracted AC from the fermentation broth with ether. The extraction rate was more than 92% [26,27]. These methods have yielded satisfactory extraction results. However, those prior studies have focused only on the extraction of AC from the fermentation broth, neither got high purity AC products nor discussed the optical purity of AC [24,25].

In our previous work [19], we investigated the AC fermentation of the BDH gene (*bdhA*) deleted strain (*BS*168D) and found that the (R)-AC content in the *BS*168D fermentation broth could reach 98% of the total AC [19]. However, a solid AC product with 99% purity obtained from the fermentation broth of *BS*168D by an efficient recovery method [26,27] revealed that the AC was a racemate, and the other 50% was (S)-AC. This finding indicated that a certain portion of (R)-AC was converted to (S)-AC during the separation process.

In this paper, we aimed to track and analyze the condition which promotes racemization as an interconversion. Then, the separation process was optimized to prevent the conversion reaction to the maximal degree and finally, the AC sample with high optical purity was obtained. To the best of our knowledge, this is the first time to discuss the optical isomer conversion during AC separation, which is important for the production of highly optically pure AC by microbial fermentation.

## 2. Materials and Methods

### 2.1. Chemicals, Stains, and Culture Media

Ethyl acetate, anhydrous methanol, and anhydrous sodium sulfate were purchased from Tianjin Kermel Chemical Reagent Co., Ltd. (Tianjin, China). *n*-Butanol was purchased from Sinopharm Chemical Reagent Co., Ltd. (Shanghai, China). All reagents were of analytical grade unless otherwise stated.

*Bacillus subtilis* strain 168D (*BS*168D) was isolated and reserved in Shandong Food Ferment Industry Research & Design Institute laboratory. The seed medium (pH 7.0–7.4) contained glucose 50 g/L, yeast extract 10 g/L, and NaCl 5 g/L, whereas the fermentation medium (pH 6.5) contained glucose 180 g/L, yeast extract 5 g/L, corn syrup 1 g/L, NaCl 5 g/L, and urea 5 g/L.

### 2.2. Preparation of AC Fermentation Broth

*Bacillus subtilis* 168D was cultured according to the method of Zhang et al. with some modifications [19]. First, *B. subtilis* 168D was pre-cultured in a 500 mL flask containing 50 mL of seed medium and incubated at 37 °C for 12 h under agitation at 200 r/min. Thereafter, the pre-culture was inoculated (10%, *v/v*) in a 5-L flask containing 800 mL of seed medium and incubated at 37 °C for 12 h under agitation at 90 r/min to obtain the secondary seed. The secondary seed was then inoculated in a 50 L fermenter containing 35 L of fermentation medium and incubated at 37 °C for 60 h under agitation at 350 r/min to obtain AC fermentation broth.

### 2.3. Extraction of AC from the Fermentation Broth

The extraction of AC from fermentation broth is based on the method of Fan et al. with some modifications [26]. The fermentation broth was filtered by a ceramic membrane to remove bacteria cells. Subsequently, the cell-free broth was evaporated with a rotary evaporator (R-100, Buchi, Switzerland) under vacuum pressure conditions (70 °C, 150 mbar), and the condensate was collected as primary AC distillate until the volume of primary AC distillate is more than 80% of the total volume of fermentation broth. Subsequently, $Na_2SO_4$ (25% *w/v*) was added to the primary AC distillate and stirred until $Na_2SO_4$ was completely dissolved. Two volumes of ethyl acetate were added to the mixture three times to extract the AC, and the AC extract solution was obtained by collecting the ethyl acetate layer after standing and partitioning.

### 2.4. Solvent Recovery from AC Extract Solution

Ethyl acetate was recovered from the AC extracts using a laboratory distillation apparatus (Figure 3), which includes a heating unit (ZNHW-2000, Lichen, China), glass distillation column, condenser, vacuum pump (R-100, Buchi, Switzerland), and collection flask. Periodically record the kettle bottom liquid temperature and take samples of condensate and kettle bottom liquid for analysis during the distillation process.

For vacuum distillation, the distillation flask was charged with AC extract, and the vacuum apparatus was switched on when the temperature of the liquid in the flask was increased to 80 °C, and the liquid temperature was maintained by adjusting the vacuum at 80 °C, 100 mabr until there was no ethyl acetate distillate. The condensate and kettle bottom liquid were regularly sampled and analyzed (with gas chromatography, Section 2.6) during the distillation process.

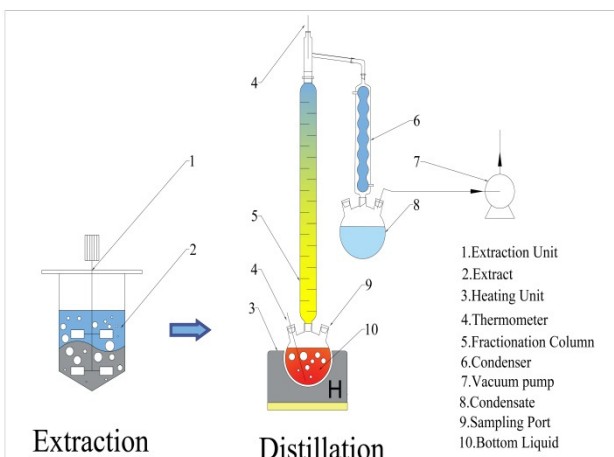

**Figure 3.** Schematic diagram of the extraction distillation unit.

### 2.5. Crystallization of AC Samples

The kettle bottom liquid obtained in Section 2.4 was incubated at 4 °C until there were crystals. The crystals were collected by filtration, washed with cold ethyl acetate, and dried at an atmospheric pressure of 20 °C to obtain AC crystals.

### 2.6. Determination of Components

All samples were diluted to a suitable concentration with anhydrous methanol. The content of AC and ethyl acetate was determined with n-butanol as the internal standard using an Agilent 7890B GC system (Agilent Technologies, CA, USA), equipped with a flame ionization detector (FID) and a Cyclodex-B chiral capillary column (30 m × 0.32 mm × 0.25 μm), which could be separated for substances with enantiomers. The temperature of both injector and detector was set at 220 °C. The hydrogen and air were 30 and 300 mL/min, respectively. The carrier gas was nitrogen and the flow rate was 1.5 mL/min. The column oven was kept at 50 °C for 5 min, then programmed to 220 °C at a rate of 30 °C/min, and finally maintained at 220 °C for 3 min. The injection volume was 1 μL. The retention times (rt) of (*R*)-AC and (*S*)-AC were 6.931 min and 7.212 min, respectively.

### 2.7. Statistical Analysis

Data were processed and plotted using Origin 2019. The results were analyzed using SPSS 17.0 software. All experiments were conducted in triplicate.

## 3. Results

### 3.1. Changes in the Ratio of AC Enantiomers in Each Separation Stage

Samples were collected from the BS168D AC fermentation broth, the primary solution, the extract solution, the kettle bottom liquid, and the AC crystals for chiral-phase GC analysis. The enantiomeric excess (e.e.) of (*R*)-AC was calculated to estimate the chiral purity of AC. The results are shown in Table 1.

**Table 1.** Ratio of (R)-AC to (S)-AC content in various separation stages. Data are given as mean ± standard deviation. Data values in a column with different superscript letters are statically different ($p \leq 0.05$).

| Separation Stage | $C_{AC}$ (g/L) | e.e. (%) | Volume (L) | pH | Recovery (%) |
|---|---|---|---|---|---|
| Fermentation broth | 45.1 ± 1.51 [b] | 93.18 ± 0.17 [a] | 5 | 4.54 ± 0.17 [a] | 100.0 ± 3.35 [a] |
| primary AC distillate | 43.2 ± 1.86 [b] | 93.16 ± 0.12 [a] | 4.7 | 3.52 ± 0.11 [b] | 90.20 ± 4.32 [b] |
| Extract | 18.8 ± 0.47 [c] | 93.14 ± 0.10 [a] | 9.3 | 3.67 ± 0.07 [b] | 77.52 ± 2.25 [c] |
| Kettle bottom liquid | 918.3 ± 37.4 [a] | 0.50 ± 0.02 [b] | 0.178 | 3.53 ± 0.12 [b] | 72.51 ± 3.04 [d] |
| AC crystal | \ | 0 ± 0.01 [b] | \ | \ | 71.26 ± 2.79 [d] |

According to Table 1, there was no significant difference in the chirality of AC in primary AC distillation and the extract solution (approximately 93%) ($p > 0.05$). However, the e.e. of (R)-AC was close to 0 in the kettle bottom liquid (0.50) ($p < 0.05$) and reached 0 in the AC crystal sample. This result suggests that the racemization occurred during the distillation stage. Therefore, the chirality change of AC during distillation was further investigated. In addition, the concentration of acetoin was 918.3 g/L after distillation (Table 1), which might be because the heating rate is too fast, resulting in rapid evaporation of AC and failure to condense in time, thus accelerating AC loss.

### 3.2. Change in the Concentration of AC Enantiomers during Distillation ee%

Firstly the ethyl acetate in the extract solution was separated by atmospheric distillation. During the distillation process, the temperature of the kettle bottom liquid gradually increased from 70 °C to 160 °C. Samples of the kettle bottom liquid were taken at different temperature points for analyzing the (R)-AC and (S)-AC content. The results showed that the content of (R)-AC and (S)-AC exhibited different trends with increasing temperatures. The e.e. of (R)-AC showed a slight change within the temperature range of 70–100 °C, a rapid decrease in the temperature range of 100 °C–140 °C, and a gradual decrease from 150 °C onwards (Figure 4). This phenomenon trend suggests that a high temperature may be the main reason caused the conversion of (R)-AC to (S)-AC.

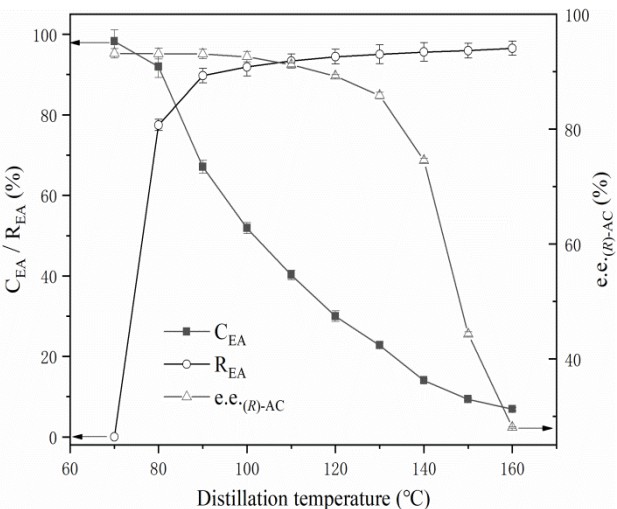

**Figure 4.** Effect of atmospheric distillation on various components. $C_{EA}$: Concentration of ethyl acetate. $R_{EA}$: Ethyl acetate recovery. e.e. $_{(R)-AC}$: Enantiomeric excess of (R)-AC.

### 3.3. Effect of Temperature on the Racemization

Samples of the fermentation broth, primary AC distillate, and the extract solution were kept warm at 60 °C, 80 °C, 90 °C, 100 °C, and 120 °C separately to estimate the effect of temperature on the racemization. As shown in Figure 5, the e.e. of (R)-AC has no significant changes held at 60–80 °C for 8 h. When the temperature was increased to 90 °C, the e.e. of (R)-AC decreased to about 92% at the end of 8 h. At 100 °C and 120 °C, the e.e. of (R)-AC decrease rate was significantly increased during the holding time. The rate of conversion between (R)-AC and (S)-AC was not significantly different among the three types of samples ($p > 0.05$), indicating that the solvents had not affected the interconversion between (R)-AC and (S)-AC. This result provides further evidence that temperature is the key factor promoting the conversion of (R)-AC and (S)-AC, with the rate of conversion increases with the increase in temperature. Under low-temperature conditions, the rate of conversion between (R)-AC and (S)-AC was significantly reduced, suggesting that the conversion of AC enantiomers may be effectively avoided by performing at low temperatures.

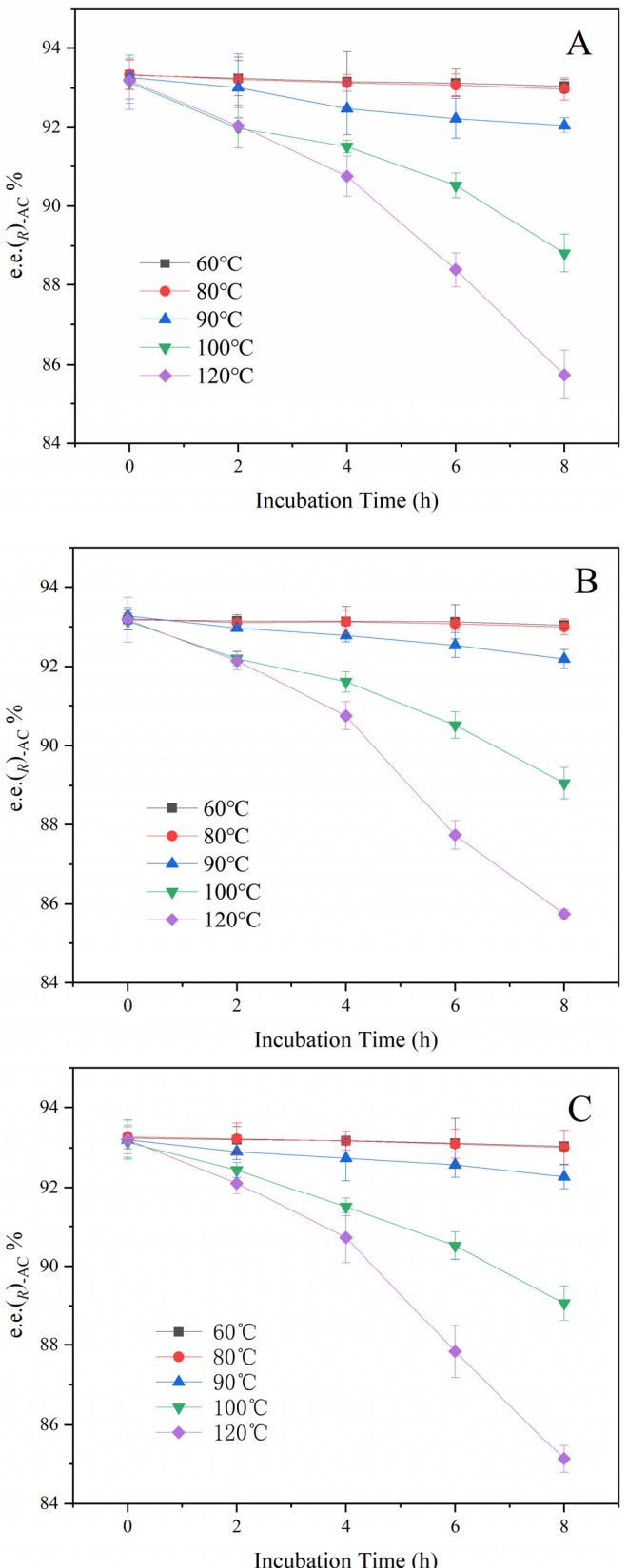

**Figure 5.** Effect of temperature on the chirality of AC. (**A**): Fermentation broth. (**B**): Primary AC distillate. (**C**): Extract.

### 3.4. Effect of pH on the Racemization

The pH of the primary AC distillate was adjusted to different values using hydrochloric acid and sodium hydroxide. Samples with different pH values were incubated at 40 °C, 60 °C, and 80 °C for 4 h to evaluate the effect of pH on the interconversion of the AC enantiomers. Figure 6 shows that the e.e. of (*R*)-AC significantly decreased at pH was <3.0 or >9.0 and that the rate of the decrease was more rapidly under alkaline conditions. The decrease in the e.e. of (*R*)-AC also positively correlated with temperature. Therefore, strong acidity and alkalinity accelerated the conversion of (*R*)-AC to (*S*)-AC and at a faster rate in alkaline environments. Although strong acidity and alkalinity promoted the conversion of (*R*)-AC to (*S*)-AC, the pH of the liquid material remained within the range of 3.5–4.5 during the various stages of AC separation and isolation (Table 1). This phenomenon suggested that pH was not the key factor responsible for the changes in e.e. of (*R*)-AC during the AC recovery process from fermentation broth.

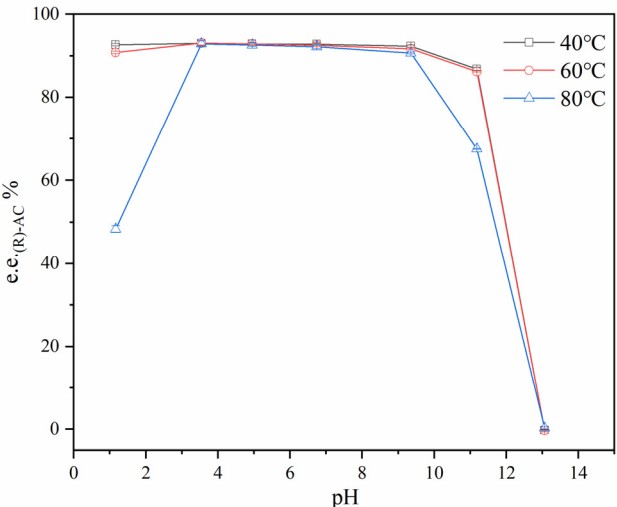

**Figure 6.** Effect of pH on the chirality of AC.

### 3.5. Optimization of the Recovery Process of Ethyl Acetate from the AC Extract Solution

Given that a high temperature is the primary cause of the conversion of (R)-AC to (S)-AC, the maintenance of an appropriate temperature during the AC recovery process will be beneficial in preserving the optical purity of AC. The experimental results described above indicate that temperatures below 80 °C favor the preservation of the optical purity of AC. Thus, the process for ethyl acetate recovery was further investigated. During atmospheric distillation, ethyl acetate began to distill when the temperature of the AC extract increased to 77 °C–78 °C. The liquid temperature gradually increased with a decrease in the ethyl acetate content of the liquid material. At 80 °C, an ethyl acetate recovery rate of up to 78% could be achieved (Figure 4); furthermore, the e.e. of (R)-AC in the liquid material remained essentially unchanged, and the AC content in the kettle bottom liquid was condensed to approximately 100 g/L and the recovery of ethyl acetate in the separation solution was reached to 78% (Figure 7). Ethyl acetate could no longer be distilled off at the atmospheric temperature when the liquid temperature was maintained at 80 °C. Thus, the system pressure had to be decreased to enable further recovery. First, vacuum rotational evaporation was used to recover the ethyl acetate from the AC extract solution at 80 °C and the results are shown in Figure 7A.

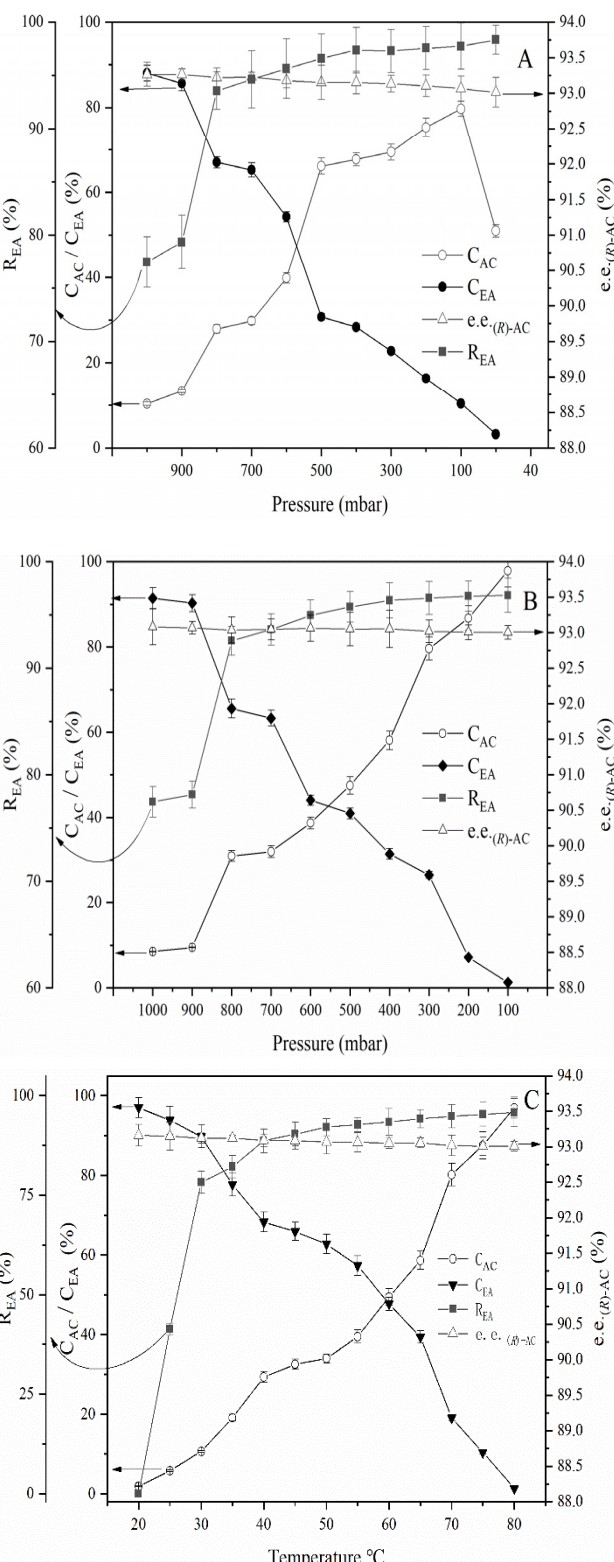

**Figure 7.** Optimization of the recovery process of ethyl acetate from the AC extract solution. (**A**): Vacuum rotational evaporation process at 80 °C. (**B**): Atmospheric and vacuum distillation combination process at 80 °C. (**C**): Vacuum distillation process (100 mbar). The horizontal axis of A and B show decreasing pressure from the left to right. $C_{AC}$: concentration of acetoin. $C_{EA}$: concentration of ethyl acetate. $R_{EA}$: ethyl acetate recovery rate. e.e. $_{(R)-AC}$: enantiomeric excess of (*R*)-AC.

It could be seen in Figure 7 the content of AC in the kettle bottom liquid gradually increased from 10.61% to 79.68% as the system pressure decreased from 1000 mbar to

100 mbar. Meanwhile, the concentration of ethyl acetate decreased to 10.13%. However, when the system pressure continued to decline to 50 mbar, the concentration of AC in the kettle bottom liquid suddenly decreased to 51.01%, and at the same time, the concentration of AC in the recovered ethyl acetate was found to increase accordingly. AC vaporized substantially when the system pressure was below 100 mbar. So ethyl acetate couldn't be fully recovered by vacuum rotational evaporation.

After atmospheric distillation was used to heat up to 80 °C in advance and the AC content in the kettle bottom liquid rose to about 100 g/L, the distillation column was kept, and the pressure was reduced to investigate the influence of atmospheric and vacuum distillation combined process on the recovery of ethyl acetate, as shown in Figure 7B. When the system pressure was decreased to 100 mbar, ethyl acetate could be fully recovered, and the purity of AC reached >98%, and the e.e. of (*R*)-AC of AC remained above 92.9%. Thus, the combined use of atmospheric and vacuum distillation enables the full recovery of ethyl acetate from the extract at lower temperatures while highly maintaining the optical purity of AC.

The AC extract solution was added to the distillation flask, the pressure of the system was adjusted by a vacuum device to 100 mbar and the heating device was opened to investigate the effect of direct vacuum distillation on the recovery of ethyl acetate. The results were shown in Figure 7C. The results showed that the kettle bottom liquid could boil at room temperature when the pressure of the system was dropped to 100 mbar, and the recovery of ethyl acetate could reach 78.02% when the temperature increased to 30 °C. When the temperature of the system rose to 80 °C, ethyl acetate could be completely recovered and the e.e. of (*R*)-AC remained above 93%, which also achieved the recovery of ethyl acetate while maintaining high chiral purity of AC.

The comparison of the three ethyl acetate recovery processes is listed in Table 2. It could be seen that the yield of AC is only 37% by the vacuum rotational evaporation at 80 °C. Although the operation time of this process is relatively short, the AC yield is very low and also the ethyl acetate could not be evaporated completely, so it is not suitable for solvent recovery. Using atmospheric and vacuum distillation combination process to separate AC and ethyl acetate, the final e.e. of (*R*)-AC of AC was 93.03%, and the operation time was 4.3 h. While if the direct vacuum distillation was used, the e.e. of (*R*)-AC of AC was 93.13, and the process took 5.6 h. To maintain the chirality of AC to the greatest extent, the direct vacuum distillation process might be the best choice to recover the ethyl acetate in the AC extract solution.

**Table 2.** Comparison of the three ethyl acetate recovery processes. $C_{AC}$: Concentration of AC. Data are given as mean ± standard deviation. Data values in a column with different superscript letters are statically different ($p \leq 0.05$).

| Separation Stage | $C_{AC}$ (g/L) | e.e. (%) | Volume (L) | Operation Time (h) | AC Yield (%) |
|---|---|---|---|---|---|
| Extract (initial conditions) | 18.8 ± 0.47 [c] | 93.14 ± 0.22 [a] | 9.3 | / | 100.0 |
| Vacuum Rotational Evaporation | 796.8 ± 18.5 [b] | 93.06 ± 0.15 [b] | 0.081 | 3.2 | 37.12 ± 1.35 [b] |
| Atmospheric + vacuum distillation | 978.5 ± 35.1 [a] | 93.03 ± 0.07 [a] | 0.171 | 4.3 | 95.93 ± 2.79 [a] |
| Direct vacuum distillation | 969.3 ± 31.7 [a] | 93.13 ± 0.08 [a] | 0.170 | 5.6 | 94.28 ± 3.21 [a] |

## 4. Discussion

AC is the simplest chiral α-hydroxy ketone that exists in two enantiomeric forms—(R)-AC and (S)-AC. Owing to its unique spatial structure, optically pure AC shows promising applications in asymmetric synthesis [3,5]. The production of optically pure AC by bacterial fermentation has recently attracted much attention. [10,23]. Currently, researchers have obtained high optical pure AC producing strains by strain screening [16], recombinant strain construction [17,18], and gene knockout [19] strategies, and the purity of optical isomers in the fermentation broth reaches more than 98%. However, compared with research on the metabolic mechanisms of bacterial strains and the optimization of fermentation processes for AC production, the downstream separation of AC from fermentation broth

is the critical step in obtaining AC products. The salting-out method [24] and aqueous two-phase extraction method [25] are several easy-to-operate methods for extracting AC from the fermentation broth, and these methods have achieved extraction rates of 90–98%. However, those researches all focused on the separation of AC from the fermentation broth, and few works is available on further purification of AC.

The (*R*)-AC yield in the fermentation broth of strain 168D, constructed in our previous work, could account for up to 98% of the total AC yield, yet only 50% of the AC crystalline product from the fermentation broth was (*R*)-AC, with the other half being (*S*)-AC. This result indicated that (*R*)-AC was converted to (*S*)-AC over the course of the separation process, and this phenomenon has never been reported before. In this paper, the high temperature was found as the main factor that promotes the conversion of (*R*)-AC to (*S*)-AC. AC is a ketone containing an α-hydroxyl group whose carbonyl group can undergo an enolization reaction similar to that of other ketones. Due to the existence of the alpha-hydroxyl group, the carbonyl enolization of AC results in the formation of 2-butene-2,3-diol [28,29], which is unstable and readily converts back to AC. However, the probability of conversion from 2-butene-2,3-diol to (*R*)—AC and (*S*)—AC is equal [30] (Figure 8). This phenomenon is greatly enhanced by high temperatures and strong acids and bases, which led to the results described above.

**Figure 8.** Mechanisms of mutarotation in AC enantiomers.

According to the results in Section 3.4, the racemization rate of AC decreases significantly when the temperature is lower than 80 °C, which provides conditions for the recovery of solvent in the extraction solution. Consequently, three processes for solvent recovery in a vacuum were compared. It was found that both atmospheric and vacuum combined distillation, as well as direct vacuum distillation, could completely recover ethyl acetate and maintain the optical purity of AC. This work discussed for the first time the changes in the optical activity of the product during the extraction of AC from the fermentation broth; compared with vacuum rotary evaporation used by Cui et al. [20] and Dai et al. [31], the vacuum distillation process is adopted in this paper to recover ethyl acetate, it is more conducive to industrial production. The operating pressure is not less than 100 mbar. The pressure conditions are relatively mild, which not only realizes the complete recovery of ethyl acetate but also maintains the optical isomer purity of AC and has a high industrial application value.

## 5. Conclusions

The optical purity of AC during the downstream separation process of AC synthesis by bio-fermentation was investigated. It was found that a high temperature during solvent recovery was a key factor promoting the conversion of (*R*)-AC to (*S*)-AC. This conversion was relatively completely preserved by controlling the temperature of the solvent recovery stage, and high-purity (*R*)-AC products were finally obtained. The methods adopted in this study provide a complete process for the production of high-chirality AC using biotechnological methods.

**Author Contributions:** Funding acquisition, J.Z.; Data curation, Y.L. and L.L.; Resources, J.L.; Supervision, J.Z., J.L. and Y.T.; Conceptualization, X.Z.; methodology, X.Z. and Z.F.; formal analysis, Z.F.; investigation, Z.F. and L.L.; writing—original draft preparation, Z.F. and X.Z.; writing—review and editing, M.Y. All authors have read and agreed to the published version of the manuscript.

**Funding:** This research was funded by Key Research and Development Project of Shandong Province (2017GSF221015), Jinan Key Research and Development Project (201866003) and Science, Education and Industry Integration project of Qilu University of Technology (2020KJC-ZD15).

**Institutional Review Board Statement:** This paper does not applicable for studies involving humans or animals.

**Informed Consent Statement:** Not applicable.

**Data Availability Statement:** This study did not report any data.

**Acknowledgments:** The authors would like to express sincere thanks to the engineers in Shandong Food Quality Supervision and Inspection Station for their kindly help on the analysis of AC.

**Conflicts of Interest:** The authors declare no conflict of interest. The funders had no role in the design of the study; in the collection, analyses, or interpretation of data; in the writing of the manuscript, or in the decision to publish the results.

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
