# Peer review of "Conversion of Enantiomers during the Separation of Acetoin from Fermentation Broth"

_fermentation, doi:10.3390/fermentation8070312_

Round 1

Reviewer 1 Report

The manuscript by Fu et al. describes and analyzes the conversion of R-acetoin to S-acetoin during the solvent distillation extraction from fermentation broth. The work will be interest to "Fermentation" readers and is appropriate for publication. However, there are several points that need clarification. The following major items need to be addressed before the manuscript can be published:

Introduction: 

The applications of acetoin isomers should be discussed. Please provide more details.

Previous works for separation and purification of pure isomer of acetoin should be added

Results:

The chemical composition of fermentation broth should be presented

The recovery yield of acetoin should be mentioned

Discussion:

 Discussion section is weak. The results obtained should be discussed thoroughly with the support of relevant references.  

Minor items:

Typing errors should be corrected e.g.  lines 15 «produced» and 275 «coulden’t»

Materials: please use italics for the scientific name of microorganism

Author Response

Response to Reviewer 1 Comments

The manuscript by Fu et al. describes and analyzes the conversion of R-acetoin to S-acetoin during the solvent distillation extraction from fermentation broth. The work will be interest to "Fermentation" readers and is appropriate for publication. However, there are several points that need clarification. The following major items need to be addressed before the manuscript can be published:

Introduction:

  1. The applications of acetoin isomers should be discussed. Please provide more details.

Response1: Thank you for your suggestions. Your suggestions are very useful to us. We have added acetoin application in the preface. Please refer to line 34 in the manuscript.

  1. Previous works for separation and purification of pure isomer of acetoin should be added

Response2: Thank you for your suggestions. Your suggestions are very meaningful. As far as we know, it is difficult to separate pure isomers from the mixture of (R)-AC and (S)-AC, so the commonly used method is to synthesize AC with high optical isomer purity from the source, and then extract AC with extractant. The main extraction methods are salting out method [1], aqueous two-phase extraction method [2]. The authors' research group added 25% Na2SO4 to the filtered fermentation broth, and extracted AC from the fermentation broth with ether. The extraction rate was over 92% [3,4]. Relevant statements have been added to the manuscript. Please refer to lines 78 -80 of the manuscript.

Results:

  1. The chemical composition of fermentation broth should be presented

Response3: Thank you very much for your suggestion. The determination methods of components in the manuscript are all completed by gas chromatography. However, the components in the fermentation broth are complex, which will cause certain damage to the gas chromatographic column. Therefore, the common method is to extract the fermentation broth with ethyl acetate and inject gas chromatography to determine the components in the fermentation broth. The results are shown in Figure S1.

Figure S1. Gas chromatogram of fermentation broh.

  1. The recovery yield of acetoin should be mentioned

Response4: thanks to your suggestion that is very beneficial to us and the AC recovery has been added to table1 and table 2. Please refer to manuscript.

Table 1. Ratio of (R)-AC to (S)-AC content in various separation stages. Data are given as mean ± standard deviation. Data values in a column with different superscript letters are statically different (p ≤  0.05).

separation stage

CAC (g/L)

e.e. (%)

volume (L)

pH

Recovery (%)

Fermentation broth

45.1±1.51b

93.18±0.17a

5

4.54±0.17a

100.0±3.35a

primary AC distillate

43.2±1.86b

93.16±0.12Ca

4.7

3.52±0.11b

90.20±4.32b

Extract

18.8±0.47c

93.14±0.10a

9.3

3.67±0.07b

77.52±2.25c

Kettle bottom liquid

918.3±37.4a

0.50±0.02b

0.178

3.53±0.12b

72.51±3.04d

AC crystal

\

0±0.01b

\

\

71.26±2.79d

Table 2. Comparison of the three ethyl acetate recovery processes. CAC: concentration of AC. Data are given as mean ± standard deviation. Data values in a column with different superscript letters are statically different (p ≤  0.05).

separation stage

CAC (g/L)

e.e. (%)

volume (L)

Operation time (h)

AC yield (%)

Extract (initial conditions)

18.8±0.47c

93.14±0.22a

9.3

/

100.0

Vacuum Rotational Evaporation

796.8±18.5b

93.06±0.15b

0.081

3.2

37.12±1.35b

Atmospheric + vacuum distillation

978.5±35.1a

93.03±0.07a

0.171

4.3

95.93±2.79a

Direct vacuum distillation

969.3±31.7a

93.13±0.08a

0.170

5.6

94.28±3.21a

Discussion:

  1. Discussion section is weak. The results obtained should be discussed thoroughly with the support of relevant references.

Response5: Thank you for your suggestions, which were helpful to us, and the results of the manuscript have been compared with other references and discussed, please refer to lines 400-405 of the manuscript.

Minor items:

  1. Typing errors should be corrected e.g. lines 15 «produced» and 275 «coulden’t»

Response6: Thank you for your suggestion, it makes sense to us, we have corrected the typing errors, please refer to line xx and line XX in the manuscript.

7 Materials: please use italics for the scientific name of microorganism

Response7: Thank you for your comments. We have revised the manuscript according to your requirements. Please refer to manuscript.

Reference

  1. Li, ; Dai, J.Y.; Xiu, Z.L. Salting-out extraction of acetoin from fermentation broths using hydroxylammonium ionic liquids as extractants, Separation and Purification Technology. 2020, 240. https://doi.org/10.1016/j.seppur.2020.116584.
  2. Sun, A.; Rao,B.; Zhang, L.Y.; Shen, Y.L.; Wei, D.Z. Extraction of acetoin from fermentation broth using an acetone/phosphate aqueous two-phase system, Chemical Engineering Communications. 2012, 199, 1492–1503. https://doi.org/10.1080/00986445.2012.683901.
  3. Liu, ; Zhao,X.; Tian,Y.; Zhang, J.; Fan, Y.; Yang, L.; Han, Y. A method for the isolation and extraction of 3-hydroxy-2-butanone from fermented broth. China Patent. 2014, CN103524315A.
  4. Liu, ; Tian, Y.; Zhao, X.; Zhang, J.; Fan, Y. A salted extraction and distillation method for the isolation of 3-hydroxy-2-butanone from fermentation broth. China Patent. 2014, CN103524315A.

Reviewer 2 Report

In this study, the authors present recovery of acetoin from fermentation broth with emphasis on maintaining the optical purity of the acetoin. Based on their results, the authors show that the primary reason for racemization during recovery is high temperature (as well as strongly acidic and basic pH), which can be mitigated for by appropriate vacuum distillation techniques (to control temperatures). The results show that the optical purity of acetoin can be effectively maintained at near R/S ratios of 28, which result from fermentation. In the absence of such control, an R/S ratio of 1 would result. The introduction of the manuscript adequately defines the scope of the work and justifies its significance. The results are presented clearly with sufficient and scientifically sound analysis. One very small suggestion is that perhaps the figures could be larger so that they would be easier to read. Please see below for a few additional remarks:

Page 2, line 50-51: Please check that butanediol has been previously defined as “BD” (I assume that’s what is meant here). It looks like only BDH as been used/defined previously, but not BD. It may also be worth double checking in general that all acronyms/abbreviations are defined appropriately at their first appearance in the text.

Figure 2: If this figure is adapted from published work in literature, please add a the relevant citation(s) in the figure caption. The layout may also benefit from a little bit more space between the end of line 61 and the top of the figure.

Section 2.3: Here you describe the extraction of acetoin, where acetoin is first in the “evaporation fluid” and then in the “primary solution.” It may be helpful or clearer to be consistent about naming/referring to the evaporation fractions by only using one name. Although probably not entirely necessary, you could consider adding an additional component to Figure 3, such that there is one image for section 2.3 and one image for section 2.4 (whereas currently the figure is only for section 2.4).

Page 4, line 128-129: I suggest to rephrase this sentence as “The condensate and kettle bottom liquid were regularly sampled and analyzed (with gas chromatography, Section 2.6) during the distillation process.” You could also consider adding information about the sampling frequency.

Figures 4 & 7-9: It may be helpful to add small arrows to this plot (particularly for ethyl acetate recovery and R/S) to show that recovery is on the left vertical axis and R/S is on the right. This may also be something to consider for Figure 7, 8 and 9. Also for Figure 7 (and fig 8), it may be helpful to briefly specify in the caption that the horizontal axis shows decreasing pressure from left to right.

Table 3: I would suggest adding one line to this table to indicate the initial conditions before evaporation/distillation. For example, you could duplicate here line 1 from Table 1 to show acetoin concentration and R/S in the fermentation broth.

Author Response

Response to Reviewer 2 Comments

  1. In this study, the authors present recovery of acetoin from fermentation broth with emphasis on maintaining the optical purity of the acetoin. Based on their results, the authors show that the primary reason for racemization during recovery is high temperature (as well as strongly acidic and basic pH), which can be mitigated for by appropriate vacuum distillation techniques (to control temperatures). The results show that the optical purity of acetoin can be effectively maintained at near R/S ratios of 28, which result from fermentation. In the absence of such control, an R/S ratio of 1 would result. The introduction of the manuscript adequately defines the scope of the work and justifies its significance. The results are presented clearly with sufficient and scientifically sound analysis. One very small suggestion is that perhaps the figures could be larger so that they would be easier to read. Please see below for a few additional remarks:

Response1: Thank you for your suggestion. Your suggestion is very helpful to us. The figure in the manuscript has been enlarged. Please refer to the manuscript for details.

  1. Page 2, line 50-51: Please check that butanediol has been previously defined as “BD” (I assume that’s what is meant here). It looks like only BDH as been used/defined previously, but not BD. It may also be worth double checking in general that all acronyms/abbreviations are defined appropriately at their first appearance in the text.

Response2: Thank you for your comment, we have reviewed the reference and 2,3-butanediol should be defined as "2,3-BD" [1,2] , thank you for correcting our error, we have revised the statement. All acronyms/abbreviations have been appropriately defined when they first appear in the text

  1. Figure 2: If this figure is adapted from published work in literature, please add a the relevant citation(s) in the figure caption. The layout may also benefit from a little bit more space between the end of line 61 and the top of the figure. Please refer to the manuscript.

Response3: Thank you for your suggestions, which are helpful to us, we have added references to figure 2 and enlarged the space between the end of line 61 and the top of the figure, Please refer to Figure 2in the manuscript.

Figure 2. Metabolic pathways of AC in organisms [23]. ALS: Acetolactate synthase; ALDC: Acetolactate decarboxylase; R-BDH: R-2.3-Butanediol dehydrogenase; meso-BDH: meso-2.3-Butanediol dehydrogenase.

  1. Section 2.3: Here you describe the extraction of acetoin, where acetoin is first in the “evaporation fluid” and then in the “primary solution.” It may be helpful or clearer to be consistent about naming/referring to the evaporation fractions by only using one name. Although probably not entirely necessary, you could consider adding an additional component to Figure 3, such that there is one image for section 2.3 and one image for section 2.4 (whereas currently the figure is only for section 2.4).

Response4: Thank you very much for your suggestions, which are very beneficial to us, and we have unified the above terms as "primary AC distillate" and an additional component had added to figure3. Please refer to section 2.3, table 1 and figure 3 of the manuscript.

Figure 3. Schematic diagram of extraction distillation unit.

  1. Page 4, line 128-129: I suggest to rephrase this sentence as “The condensate and kettle bottom liquid were regularly sampled and analyzed (with gas chromatography, Section 2.6) during the distillation process.” You could also consider adding information about the sampling frequency.

Response5: Thank you for your suggestions, which are helpful for the improvement of the quality of our manuscript. We have modified lines 128-129 according to your suggestions. Please refer to lines 146-148 of the manuscript.

  1. Figures 4 & 7-9: It may be helpful to add small arrows to this plot (particularly for ethyl acetate recovery and R/S) to show that recovery is on the left vertical axis and R/S is on the right. This may also be something to consider for Figure 7, 8 and 9. Also for Figure 7 (and fig 8), it may be helpful to briefly specify in the caption that the horizontal axis shows decreasing pressure from left to right.

Response6: Thank you very much for your suggestions, which are very helpful to us. Small arrows have been added to figures 4 and 7-9. The decreasing pressure from left to right shown on the horizontal axis has been added in figures 7 and 8. Figures 7-9 has been integrated into a new figure 7. Please refer to figures 4 and 7 in the manuscript.

Figure 7. Optimization the recovery process of ethyl acetate from the AC extract solution. A: Vacuum rotational evaporation process at 80℃. B: Atmospheric and vacuum distillation combination process at 80℃. C: Vacuum distillation process (100mbar). The horizontal axis of A and B shows decreasing pressure from left to right. CAC: concentration of acetoin. CEA: concentration of ethyl acetate. REA: ethyl acetate recovery rate. e.e.(R)-AC: enantiomeric excess of (R)-AC.

  1. Table 3: I would suggest adding one line to this table to indicate the initial conditions before evaporation/distillation. For example, you could duplicate here line 1 from Table 1 to show acetoin concentration and R/S in the fermentation broth.

Response7: Thank you for your suggestion. The initial conditions before evaporation / distillation have been added to table 3. Please refer to table 2 in the manuscript.

Table 2. Comparison of the three ethyl acetate recovery processes. CAC: concentration of AC.

Data are given as mean ± standard deviation. Data values in a column with different superscript letters are statically different (p ≤  0.05).

separation stage

CAC (g/L)

e.e. (%)

volume (L)

Operation time (h)

AC yield (%)

Extract (initial conditions)

18.8±0.47c

93.14±0.22a

9.3

/

100.0

Vacuum Rotational Evaporation

796.8±18.5b

93.06±0.15b

0.081

3.2

37.12±1.35b

Atmospheric + vacuum distillation

978.5±35.1a

93.03±0.07a

0.171

4.3

95.93±2.79a

Direct vacuum distillation

969.3±31.7a

93.13±0.08a

0.170

5.6

94.28±3.21a

Reference

  1. Xiu J. 2011. Salting-out Extraction of 2,3-Butanediol from Jerusalem artichoke-based Fermentation Broth. Chinese Journal of Chemical Engineering.
  2. Svenja, Kochius, Melanie, Paetzold, Alexander, Scholz, Hedda, Merkens, Andreas, Vogel. Enantioselective enzymatic synthesis of the α-hydroxy ketone (R)-acetoin from meso-2,3-butanediol, Journal of Molecular Catalysis B Enzymatic. 2014, 103, 61-66. https://doi.org/10.1016/j.molcatb.2013.08.016

Reviewer 3 Report

Comments to Author(s) of Manuscript : 

Manuscript Number: fermentation-1759668

Mechanisms of enantiomeric conversion during the separation of acetoin from fermentation broth

Journal: Fermentation

This manuscript showed optimization of the downstream purification process to maintain the chirality of acetoin produced from microbial production, which is mainly composed of (R)-acetoin (AC). The authors i) identified that the additional distillation step after salting-out increased the racemization, ii) searched for the factors (temperature and pH) that reduce the chirality of AC, and iii) evaluated three distillation methods for increased ethyl acetate recovery. The data show that temperature was a critical factor to prevent any racemization of AC in the downstream process.

            The manuscript has a straightforward goal to achieve reducing the racemization of R-AC during the downstream process. Readers of "Fermentation" will find it interesting and informative as the manuscript highlights the importance of downstream process optimization. However, the sentences and the overall structure are not easy to understand the message the authors want to address. There should be a significant amount of effort put into this manuscript to clarify the sentences. There are lots of errors in sentences with missing subjects or misuse of passive/active voice.

1.     The word “Mechanism” in the title does not fit with the manuscript. This manuscript only discussed the optimization of R-AC separation process by showing the effects of temperature, pH, and pressure in R-AC/S-AC and its yield. If it were to talk about the mechanism of enantiomeric conversion, the authors should prove S-AC separation process also shows the same trend in temperature, pH, and pressure, detect any intermediate during racemization, or show kinetic analysis of how temperature affected the racemization rate. The reviewer suggests that the authors modify the title to clarify that the manuscript is about optimization. 

2.     “Given that AC is an intermediate product of the metabo- 41 lism of sugars in several organisms, so it could be produced by fermentation [9]. There- 42 fore, AC with high optical purity might be produced using biotechnology.” 

Explanation of why biological production of AC outperforms chemical synthesis is not smooth. 

3.     In the “introduction” section, the authors take a large part in describing the biological production of R-AC. since the authors focus on the downstream rather than the biological production throughout the manuscript, the “Introduction” should be reorganized. The reviewer recommends the authors summarize the upstream process along with the importance of acetoin and thoroughly introduce previous studies in acetoin downstream process or purification process. For the same reason, the authors should consider changing Figure 2 to a scheme of the whole separation process. For every step, there is lacking information including the absolute amount of R-AC. For instance, the data or graph shown in this manuscript is not able to infer the scale of the purification. Starting concentration and volume of distillation should be noted.  

4.     “The downstream isolation of AC is important after optimising the metabolic mech- 66 anism of microorganisms and fermentation conditions. Salting-out [21] and aqueous 67 two-phase extraction [22] are currently the main methods used to extract AC from fer- 68 mentation broth, these methods have yielded satisfactory extraction results. However, 69 those prior studies have focused only on the extraction of AC from fermentation broth, 70 neither got high purity AC products nor discussed the optical purity of AC [21,22].”

The authors cited patents in the manuscript, which is quite difficult to track the data for the reader. The reviewer recommends specifying the values for better insights into how much improvement this manuscript has achieved compared to the previous cases.

5.     “This is the first time to 82 discuss the optical isomer conversion during AC separation,”

“AC separation” could include “AC extraction from the fermentation broth", which was already successful in the previous result of your group. Would this be “AC purification”?

6.     “Subsequently, the broth free of cell(s) was evaporated with rotary 108 evaporator (R-100, Buchi, Switzerland) under vacuum pressure conditions (70 °C, 150 109 mbar), and the evaporation fluid was collected as primary AC solution.”

The sentence sounds like the authors evaporated until there is no liquid remaining in the round-bottom flask. It would be great if the authors can clarify the sentences throughout the methods section.

7.     The section “3.2 Effect of distillation …” showed that the increasing temperature decreases the R/S ratio, proving the “effect of temperature” on the content of R-AC and S-AC. Since there are no variables other than temperature, this section should be combined with “3.3 Effect of temperature on the conversion of R-AC to S-AC”

8.      The authors used “primary solution” or “primary AC solution” in the “materials and methods” section, while they used “primary AC distillate” in the “results” section. The reviewer recommends the authors use the same words for the sample type throughout the manuscript.

9.     Table 1 and Table 2 are combined into one table, which might help readers to understand the changes in factors for each separation stage. Additionally, all the tables have “a” or “b” that are not explained in the caption. Please add an explanation of what the superscript indicates in the table. 

10.  “The experimental results de- 224 scribed above indicate that temperatures below 80 °C favour the preservation of the op- 225 tical purity of AC. Thus, the process for ethyl acetate recovery was further investigated. “

The reviewer is not convinced why ethyl acetate recovery is the problem since the boiling point of ethyl acetate is below 80’C. There should be a clarified explanation why the authors look into different distillation methods to increase the ethyl acetate recovery or validate the amount of ethyl acetate recovery.

11.  Throughout the distillation experiments, the reviewer is curious if the distillation time was identical for all of the data points. If the distillation time was identical, the distillation time should be indicated in the caption. If the graph is showing one set of distillation experiments, the figures should show the pressure/temperature changes according to time and more thorough descriptions are required of how the experiments have been performed.

12.  In figure 7-9, the initial concentration of AC looks different for three distillation experiments. The authors should indicate the initial volume and concentration of the AC extract solution used in the distillation experiments. Additionally, it would be easier to understand the figures if the three figures are combined into one.

13.  The authors are taking a large portion of the “discussion” section to discuss the upstream of R-AC production. The reviewer agrees to emphasize the fact that most studies focus on achieving high optical purity in fermentation. However, it is not recommended to repeat the description in the “introduction” and dominate the discussion about the result that has been actually done in this manuscript. 

14.  The authors showed that the R/S value after the crystallization of AC was about 1 when the distillation process was not optimized. To make a fair comparison and prove that crystallization does not affect racemization, it is necessary to compare the R/S value of the final product. The reviewer suggesting to share R/S value of the AC crystal produced from the current distillation process (or generating the current process version of table 1) to highlight the impressive achievement of this manuscript.

Author Response

Response to Reviewer 3 Comments

This manuscript showed optimization of the downstream purification process to maintain the chirality of acetoin produced from microbial production, which is mainly composed of (R)-acetoin (AC). The authors i) identified that the additional distillation step after salting-out increased the racemization, ii) searched for the factors (temperature and pH) that reduce the chirality of AC, and iii) evaluated three distillation methods for increased ethyl acetate recovery. The data show that temperature was a critical factor to prevent any racemization of AC in the downstream process.

The manuscript has a straightforward goal to achieve reducing the racemization of R-AC during the downstream process. Readers of "Fermentation" will find it interesting and informative as the manuscript highlights the importance of downstream process optimization. However, the sentences and the overall structure are not easy to understand the message the authors want to address. There should be a significant amount of effort put into this manuscript to clarify the sentences. There are lots of errors in sentences with missing subjects or misuse of passive/active voice.

Response: Thank you very much for pointing out our shortcomings. Your suggestions are very helpful to us. We have corrected the relevant grammatical errors in the manuscript. Please refer to the manuscript for details.

  1. The word “Mechanism” in the title does not fit with the manuscript. This manuscript only discussed the optimization of R-AC separation process by showing the effects of temperature, pH, and pressure in R-AC/S-AC and its yield. If it were to talk about the mechanism of enantiomeric conversion, the authors should prove S-AC separation process also shows the same trend in temperature, pH, and pressure, detect any intermediate during racemization, or show kinetic analysis of how temperature affected the racemization rate. The reviewer suggests that the authors modify the title to clarify that the manuscript is about optimization.

Response1: Thank you very much for your suggestions. We believe that your suggestions are very reasonable after carefully reading the manuscript. The word "mechanism" may not be suitable for this article, so we revised the title of the manuscript to “Conversion of enantiomers during the separation of acetoin from fermentation broth”.

  1. “Given that AC is an intermediate product of the metabo- 41 lism of sugars in several organisms, so it could be produced by fermentation [9]. There- 42 fore, AC with high optical purity might be produced using biotechnology.”

Explanation of why biological production of AC outperforms chemical synthesis is not smooth.

Response2: Thank you very much for pointing out the inadequacies, which helped us to follow up, the related statement in the manuscript has been revised. Please refer to line 44-46 in the manuscript.

  1. In the “introduction” section, the authors take a large part in describing the biological production of R-AC. since the authors focus on the downstream rather than the biological production throughout the manuscript, the “Introduction” should be reorganized. The reviewer recommends the authors summarize the upstream process along with the importance of acetoin and thoroughly introduce previous studies in acetoin downstream process or purification process. For the same reason, the authors should consider changing Figure 2 to a scheme of the whole separation process. For every step, there is lacking information including the absolute amount of R-AC. For instance, the data or graph shown in this manuscript is not able to infer the scale of the purification. Starting concentration and volume of distillation should be noted.

Response3: Thank you very much for your suggestion, which we think is very relevant. After careful analysis of the manuscript, we believe that the source of the series of isolation processes used in this paper comes from genetically modified strains, which is the most fundamental feature of this work. The downstream isolation process of AC is an important part of obtaining the AC product and both must be taken into account to highlight the novelty of this work. Therefore, we consider that the statement in the introduction regarding the metabolism and modification of the strain should be retained. In addition, the AC content in each isolation stage has been added in Tables 1 and 2. Please Tables 1 and 2 in the manuscript.

Table 1. Ratio of (R)-AC to (S)-AC content in various separation stages. Data are given as mean ± standard deviation. Data values in a column with different superscript letters are statically different (p ≤  0.05).

separation stage

CAC (g/L)

e.e. (%)

volume (L)

pH

Recovery (%)

Fermentation broth

45.1±1.51b

93.18±0.17a

5

4.54±0.17a

100.0±3.35a

primary AC distillate

43.2±1.86b

93.16±0.12Ca

4.7

3.52±0.11b

90.20±4.32b

Extract

18.8±0.47c

93.14±0.10a

9.3

3.67±0.07b

77.52±2.25c

Kettle bottom liquid

918.3±37.4a

0.50±0.02b

0.178

3.53±0.12b

72.51±3.04d

AC crystal

\

0±0.01b

\

\

71.26±2.79d

Table 2. Comparison of the three ethyl acetate recovery processes. CAC: concentration of AC. Data are given as mean ± standard deviation. Data values in a column with different superscript letters are statically different (p ≤  0.05).

separation stage

CAC (g/L)

e.e. (%)

volume (L)

Operation time (h)

AC yield (%)

Extract (initial conditions)

18.8±0.47c

93.14±0.22a

9.3

/

100.0

Vacuum Rotational Evaporation

796.8±18.5b

93.06±0.15b

0.081

3.2

37.12±1.35b

Atmospheric + vacuum distillation

978.5±35.1a

93.03±0.07a

0.171

4.3

95.93±2.79a

Direct vacuum distillation

969.3±31.7a

93.13±0.08a

0.170

5.6

94.28±3.21a

  1. “The downstream isolation of AC is important after optimising the metabolic mech- 66 anism of microorganisms and fermentation conditions. Salting-out [1] and aqueous 67 two-phase extraction [2] are currently the main methods used to extract AC from fer- 68 mentation broth, t22hese methods have yielded satisfactory extraction results. However, 69 those prior studies have focused only on the extraction of AC from fermentation broth, 70 neither got high purity AC products nor discussed the optical purity of AC [3,4].”

The authors cited patents in the manuscript, which is quite difficult to track the data for the reader. The reviewer recommends specifying the values for better insights into how much improvement this manuscript has achieved compared to the previous cases.

Response4: Thank you for your suggestion. The critical data mentioned in the patent has been added in the manuscript. Please refer to lines 78 -80 of the manuscript.

  1. “This is the first time to 82 discuss the optical isomer conversion during AC separation,”

“AC separation” could include “AC extraction from the fermentation broth", which was already successful in the previous result of your group. Would this be “AC purification”?

Response5: Thank you very much for your comments, they were very helpful. In our previous work, we carried out early purification work, i.e. fermentation, filtration, extraction, distillation and crystallisation to obtain a highly pure AC product, providing a complete process for the biotechnological production of AC, but did not focus on the conversion of AC enantiomers during the separation process. The work done in this paper is an optimisation of the earlier work to provide a process route for the production of pure AC with high enantiomers.

  1. “Subsequently, the broth free of cell(s) was evaporated with rotary 108 evaporator (R-100, Buchi, Switzerland) under vacuum pressure conditions (70 °C, 150 109 mbar), and the evaporation fluid was collected as primary AC solution.” The sentence sounds like the authors evaporated until there is no liquid remaining in the round-bottom flask. It would be great if the authors can clarify the sentences throughout the methods section.

Response6: Thank you very much for pointing out the inadequacy of the manuscript, "until the volume of "primary AC distillate" is more than 80% of the total volume of fermentation broth" which as a condition to judge the end of evaporation has been added after "collected as the primary AC solution. " Please refer to lines 125-126 in the manuscript.

  1. The section “3.2 Effect of distillation …” showed that the increasing temperature decreases the R/S ratio, proving the “effect of temperature” on the content of R-AC and S-AC. Since there are no variables other than temperature, this section should be combined with “3.3 Effect of temperature on the conversion of R-AC to S-AC”

Response7: Thank you very much for your comments. We think what you said is very reasonable. However, after careful consideration of the experimental results in sections 3.2 and 3.3, we believe that there are still differences between the two. First, the experiment in Section 3.2 is a process of continuous heating up. With the continuous evaporation of ethyl acetate, the concentration of ethyl acetate in the bottom liquid is continuously reduced. Therefore, the time interval between each temperature node is different. In Section 3.3, the effect of temperature on the proportion of AC racemization was investigated at the same time interval. Therefore, we believe that it is more reasonable to discuss them separately.

  1. The authors used “primary solution” or “primary AC solution” in the “materials and methods” section, while they used “primary AC distillate” in the “results” section. The reviewer recommends the authors use the same words for the sample type throughout the manuscript.

Response8: Thank you for your comments. Your comments are very important to us. We have unified the above terms as "primary ac disassemble" Please refer to the manuscript.

  1. Table 1 and Table 2 are combined into one table, which might help readers to understand the changes in factors for each separation stage. Additionally, all the tables have “a” or “b” that are not explained in the caption. Please add an explanation of what the superscript indicates in the table.

Response9: Thanks for your suggestions, your suggestions are very meaningful, tables 1 and 2 have been combined into a new table 1, all tables " a " or " b " have been reinterpreted, please refer to tables 1 and 2 in the manuscript.

  1. “The experimental results de- 224 scribed above indicate that temperatures below 80 °C favour the preservation of the op- 225 tical purity of AC. Thus, the process for ethyl acetate recovery was further investigated. “

The reviewer is not convinced why ethyl acetate recovery is the problem since the boiling point of ethyl acetate is below 80’C. There should be a clarified explanation why the authors look into different distillation methods to increase the ethyl acetate recovery or validate the amount of ethyl acetate recovery.

Response10: Thank you for your comments. Your comments are very important to us. When ethyl acetate is a pure substance, its boiling point is 77 ℃ under normal pressure. However, when ethyl acetate forms a solution with other substances with high boiling points, the boiling point of the feed liquid will increase with the distillation of ethyl acetate. In particular, too fast heating may cause insufficient evaporation of ethyl acetate (chapter 3.2), resulting in poor distillation effect. This is also the negligence of our previous distillation operation. Therefore, in the experiment of optimizing solvent recovery in this paper (Chapter 3.5), the gradient heating method is adopted to ensure the full recovery of ethyl acetate.

Since the content of AC is still very low after the extraction operation, ethyl acetate needs to be removed by distillation. In addition, according to chapter 3.3, when the temperature is <80 ℃, the racemization rate of AC is slow. Therefore, in order to obtain high-purity AC and maintain the proportion of optical isomers of AC, different distillation methods are adopted in Chapter 3.5.

  1. Throughout the distillation experiments, the reviewer is curious if the distillation time was identical for all of the data points. If the distillation time was identical, the distillation time should be indicated in the caption. If the graph is showing one set of distillation experiments, the figures should show the pressure/temperature changes according to time and more thorough descriptions are required of how the experiments have been performed.

Response11: Thank you for your comments. In the whole distillation experiment, since ethyl acetate was constantly distilled out of the kettle bottom liquid, resulting in a constantly decreasing ethyl acetate concentration in the kettle bottom liquid. Hence, the distillation time was not the same for all data points.

  1. In figure 7-9, the initial concentration of AC looks different for three distillation experiments. The authors should indicate the initial volume and concentration of the AC extract solution used in the distillation experiments. Additionally, it would be easier to understand the figures if the three figures are combined into one.

Response12: Thank you very much for your comments. The AC extract used in sections 3.5.1 and 3.5.2 is heated to 80 ℃ through atmospheric distillation in advance to remove about 78% of ethyl acetate, and then kept at 80 ℃ to gradually reduce the system pressure. However, in Chapter 3.3.3, AC extraction liquid without frequent pressure rectification is directly used. Therefore, when the pressure drops to 100mbar, only a little heating is required to make the feed liquid boil. The initial concentration of AC in the process optimization experiment has been indicated in the manuscript. Please refer to lines 268-269, 303-304 of the manuscript for details. In addition, we have combined into figure 7-9 into one. Please refer to the manuscript.

Figure 7. Optimization the recovery process of ethyl acetate from the AC extract solution. A: Vacuum rotational evaporation process at 80℃. B: Atmospheric and vacuum distillation combination process at 80℃. C: Vacuum distillation process (100mbar). The horizontal axis of A and B shows decreasing pressure from left to right. CAC: concentration of acetoin. CEA: concentration of ethyl acetate. REA: ethyl acetate recovery rate. e.e.(R)-AC: enantiomeric excess of (R)-AC

  1. The authors are taking a large portion of the “discussion” section to discuss the upstream of R-AC production. The reviewer agrees to emphasize the fact that most studies focus on achieving high optical purity in fermentation. However, it is not recommended to repeat the description in the “introduction” and dominate the discussion about the result that has been actually done in this manuscript.

Response13: Thank you for your suggestions. Your suggestions are very helpful to improve the quality of the manuscript. We have simplified the discussion and added the comparison between the experimental results of this paper and other references. Please refer to lines 351-352, 359-361, 367-368,392-394, 400-405 of the manuscript.

  1. The authors showed that the R/S value after the crystallization of AC was about 1 when the distillation process was not optimized. To make a fair comparison and prove that crystallization does not affect racemization, it is necessary to compare the R/S value of the final product. The reviewer suggesting to share R/S value of the AC crystal produced from the current distillation process (or generating the current process version of table 1) to highlight the impressive achievement of this manuscript.

Response14: Thank you very much for your suggestion, which is of great interest to us. After the optimization of the separation process, the AC product obtained was also placed at 4°C to observe its crystallization. The results show that pure AC with high optical isomer has no crystallization, probably because the single configuration of AC could not combine with the AC of another configuration to form a dimer. Therefore, AC with high optical isomer content obtained after vacuum distillation can be the final product.

Reviewer 4 Report

The manuscript “Mechanisms of enantiomeric conversion during the separation of acetoin from fermentation broth” describe the results of a research on the effect of temperature on the optical purity of acetoin recovered from the fermentation broth through several steps:

 distillation of the filtered broth affords an aqueous solution called primary distillate by the Authors,

 the extraction of the primary distillate with ethyl acetate affords am organic solution called AC extraction solution, which is subsequently distillated to recover ethyl acetate (distillate) and acetoin (as the residue in the boiler).

crystallization from the residual aqueous solution in the boiler of the first distillation.

The reported results demonstrate the positive effect of high temperature on the racemization of acetoin. The Authors demonstrated that, the switching from atmospheric distillation to a vacuum (100 mbar) distillation can avoid racemization by lowering the bp of the solvents.  

These are a too predictable results for anyone who has a little of practice in the chemistry laboratory. In the same way, an ordinary chemist knows that basic and acidic conditions promote alpha-hydroxyketones racemization through the transient formation of the enediol intermediate, as well as that, such most of the chemical transformation, the racemization is accelerated by temperature increasing.

The Authors gave a long description of the trend of the solvent recovery by evaporation as a function of the vacuum and temperature applied to the distillation apparatus. These are data which can be easily obtained simply consulting a pressure-temperature normograph (an on-line version has been suggested to the Authors in the attached correction file).

None has been written, at least as a hypothesis, on chemical species present in the filtered broth, in the primary distillate and in the organic extract as well, which could promote the racemization (volatile organic acids or amines?).    

Furthermore, the manuscript contains several conceptual mistake, following the main ones

The Authors wrongly used the term “chirality”, acetoin is a chiral compound and it does not become an achiral one! so, chirality does not change. What that changes during the distillation is the enantiomeric composition of the acetoin (enantiopurity or optical purity) which is normaly defined by the enatiomeric excess which in my opinion is more correct than the enantiomeric ratio (R)/(S) used by the Authors, A definition of the enantiomeric excess (e.e.%) has been given in the attached correction file.

In the manuscript the Authors refers to the racemization as a “conversion of (R)-AC to (S)-AC. This is uncorrected because what happen is an interconversion.

Many other mistakes and inaccuracies have been highlighted by the referee in the attached pdf file.  

Author Response

Response to Reviewer 4 Comments

The manuscript “Mechanisms of enantiomeric conversion during the separation of acetoin from fermentation broth” describe the results of a research on the effect of temperature on the optical purity of acetoin recovered from the fermentation broth through several steps:

 distillation of the filtered broth affords an aqueous solution called primary distillate by the Authors,

 the extraction of the primary distillate with ethyl acetate affords am organic solution called AC extraction solution, which is subsequently distillated to recover ethyl acetate (distillate) and acetoin (as the residue in the boiler).

crystallization from the residual aqueous solution in the boiler of the first distillation.

The reported results demonstrate the positive effect of high temperature on the racemization of acetoin. The Authors demonstrated that, the switching from atmospheric distillation to a vacuum (100 mbar) distillation can avoid racemization by lowering the bp of the solvents. 

These are a too predictable results for anyone who has a little of practice in the chemistry laboratory. In the same way, an ordinary chemist knows that basic and acidic conditions promote alpha-hydroxyketones racemization through the transient formation of the enediol intermediate, as well as that, such most of the chemical transformation, the racemization is accelerated by temperature increasing.

Response1: Thank you very much for your comments, we agree with you very much, and our experimental results also proved that the temperature could improve the racemization rate, on the basis of this, we experimentally quantified the determination of the temperature effect on the content of AC racemization at a certain time, and the results showed that the racemization rate of AC would decrease obviously when the temperature was lower than 80 ℃, which provided the conditions for the process optimization afterwards, More after scale-up provides process guidance.

The Authors gave a long description of the trend of the solvent recovery by evaporation as a function of the vacuum and temperature applied to the distillation apparatus. These are data which can be easily obtained simply consulting a pressure-temperature normograph (an on-line version has been suggested to the Authors in the attached correction file).

Response2: Thank you for your comments. Your comments are very meaningful to us. Before solvent recovery, we also predicted the binary system composed of AC and ethyl acetate through simulation software (Aspen Plus) (Fig. S2), and found that AC and ethyl acetate had no azeotropic point within the selected pressure (30mbar, 50mbar and 100mbar). When the pressure is 100mbar and the temperature is 80 ℃, the content of AC is 82.5%. However, in the real vacuum distillation experiment, when the pressure is 100mbar and the temperature is 80 ℃, the content of AC has reached more than 96% (Fig. 8.and Fig. 9.), This shows that there is still some error between the prediction results and the real experimental data. In the subsequent work, we will continue to optimize the calculation results according to the experimental results.

Figure S2. T-xy phase diagram of AC with ethyl acetate

Figure 7-B. Atmospheric and vacuum distillation combination process at 80℃. The horizontal axis shows decreasing pressure from left to right.CAC: concentration of acetoin. CEA: concentration of ethyl acetate. REA: ethyl acetate recovery rate. e.e.(R)-AC: enantiomeric excess of (R)-AC.

Fihure 7-C. Vacuum distillation process (100mbar). CAC: concentration of acetoin. CEA: concentration of ethyl acetate. REA: ethyl acetate recovery rate. e.e.(R)-AC: enantiomeric excess of (R)-AC.

None has been written, at least as a hypothesis, on chemical species present in the filtered broth, in the primary distillate and in the organic extract as well, which could promote the racemization (volatile organic acids or amines?).

Response3: Thank you very much for your comments. We think your comments are very correct. When analyzing the cause of AC racemization, we suspected that some substance might promote the racemization of AC, and then analyzed the components in the fermentation broth by gas chromatography (Fig. S1). It was found that the content of other substances except AC was not high, so we guessed that temperature was the main factor affecting the conversion of optical isomers in the process of AC separation, but this was not consistent with the title of this paper, Therefore, we revised the title of the manuscript to "conversion of enablers during the separation of acid from induction broth". Please refer to the manuscript.

Figure S1. Gas chromatogram of fermentation broh.

Furthermore, the manuscript contains several conceptual mistake, following the main ones

The Authors wrongly used the term “chirality”, acetoin is a chiral compound and it does not become an achiral one! so, chirality does not change. What that changes during the distillation is the enantiomeric composition of the acetoin (enantiopurity or optical purity) which is normaly defined by the enatiomeric excess which in my opinion is more correct than the enantiomeric ratio (R)/(S) used by the Authors, A definition of the enantiomeric excess (e.e.%) has been given in the attached correction file.

Response4: Thank you very much for your suggestion, we think it makes a lot of sense and we have revised the R/S to ee%. Please refer to the manuscript.

In the manuscript the Authors refers to the racemization as a “conversion of (R)-AC to (S)-AC. This is uncorrected because what happen is an interconversion.

Response5: Thank you for your comment, it is very important to us and we have changed "conversion of (R)-AC to (S)-AC" to "racemization" as required by you. Please refer to the manuscript.

Many other mistakes and inaccuracies have been highlighted by the referee in the attached pdf file. 

Response6: Thank you for your comments, it was very significant to us and we have corrected other errors and inaccuracies in this article as you requested. Please refer to the manuscript for details.

Round 2

Reviewer 1 Report

The authors have, in my opinion, satisfactorily addressed the issues raised by myself and the other reviewers, hence the manuscript is appropriate for publication. 

Author Response

Response to Reviewer 1 Comments

Thank you very much for taking time out of your busy schedule to judge our response letter and manuscript, and you are very grateful for your evaluation of the revised version of our manuscript. We will now submit our manuscript, which has been accepted for revision, as follows.
